# Integrated Analysis by GC/MS and $^{13}$C NMR of Moroccan *Cladanthus mixtus* Essential Oil; Identification of Uncommon Epoxyfarnesanes

Souad El Hafidi [1,2], Khadija Bakhy [2], Mohammed Ouhssine [1], Abderrahim Benzakour [1], Joseph Casanova [3], Mathieu Paoli [3] and Félix Tomi [3,*]

1 Laboratory of Natural Resources and Sustainable Development, Faculty of Science, University Ibn Tofail, BP 242, Kenitra 14000, Morocco; elhafidisoad@gmail.com (S.E.H.); ouhssineunivit@gmail.com (M.O.); abderrahim.benzakour@uit.ac.ma (A.B.)
2 Research Unit on Aromatic and Medicinal Plant, INRA, Rabat-Instituts, BP 6570, Rabat 10101, Morocco; bakhy2@gmail.com
3 Laboratoire Sciences Pour l'Environnement, Université de Corse-CNRS, UMR 6134 SPE, Route des sanguinaires, 20000 Ajaccio, France; joseph.casanova@wanadoo.fr (J.C.); paoli_m@univ-corse.fr (M.P.)
* Correspondence: tomi_f@univ-corse.fr

**Abstract:** *Cladanthus mixtus* (L.) Chevall., Asteraceae, also known as Moroccan chamomile, is a spontaneous, annual plant growing wild in North-Western Morocco. Economically, the essential oil of *C. mixtus* is of high interest, Morocco being the only supplier on the international market. Two essential oil samples (EO) were isolated from aerial parts of *Cladanthus mixtus* (L.) Chevall., and analyzed by a combination of chromatographic and spectroscopic techniques (gas chromatography (GC) in combination with retention indices (RI), gas chromatography-mass spectrometry (GC/MS), and $^{13}$C NMR spectroscopy). Computer matching against the in-house $^{13}$C NMR library allowed the identification of the eight components at appreciable contents, namely 3,6,6,9-bis-epoxy-farnesa-1,7(14),10-triene, and its 3-epi, 9-epi, and 3,9-diepi epimers, and 6,9-epoxy-farnesa-1,7(14),10-trien-3-ol and its 3-epi, 6-epi, and 3,6-diepi epimers. Our results confirm the tremendous chemical variability of Moroccan *C. mixtus* essential oil and the usefulness of $^{13}$C NMR analysis, in combination with GC(RI), for the identification of uncommon oxygenated sesquiterpenes that induce an original composition.

**Keywords:** *Cladanthus mixtus*; chemical variability; bis-epoxy-farnesa-1,7(14),10-triene; 6,9-epoxy-farnesa-1,7(14),10-trien-3-ol; $^{13}$C NMR



## 1. Introduction

*Cladanthus mixtus* (L.) Chevall. (synonyms: *Anthemis mixta* (L.), *Chamaemelum mixtum* (L.) All., *Ormenis mixta* (L.) Dumort, *Ormenis multicaulis* Braun-Blanq. & Maire), also known as Moroccan chamomile, Asteraceae, is a spontaneous, annual plant 10 to 40 cm tall with numerous erect, lying, or ascending stems terminated by fragrant flower heads with ligulate, white, and sterile external flowers decorated with yellow at their base and fertile yellow tuberous internal flowers. This species is a sialophyte that abounds in the voids of semi-arid and sub-humid bioclimates on sandy soils of the thermomediterranean stage. It is generally found in open forests, fields, sand and stone agricultural landscapes, and low mountains. In Morocco, this plant is known by the vernacular name "Hellâla" [1] and is mainly distributed in two disjointed areas, the first between Tangier, Ouezzane, Souk Larbaa, Moulay Bousselham, and Azilah, and the second between Kenitra, Sidi Slimane, Khémisset, and Rabat [2]. Economically, the essential oil of *C. mixtus* is of high interest. Morocco is the world's leading producer [3]. In terms of quality, the essential oil of *C. mixtus* was ranked ninth among the 20 best essential oils produced in Morocco. In Morocco, *C. mixtus* is advised as an anxiolytic for the rebalancing of the central nervous system; it has great value in nervous breakdowns and for mild hepatic and gastric insufficiency and

*Colibacillary colitis* [4]. Thanks to its pleasant smell, the essential oil of *C. mixtus* is sought after in perfumery, cosmetics, and medicine [4]. Concerning the chemical composition of the essential oil of *Cladanthus mixtus* (Moroccan chamomile), various studies have shown a very important chemical polymorphism.

The chemical composition of *C. mixtus* essential oil has been investigated, most of the time, by GC/MS in combination with retention indices on the non-polar or semi-polar chromatography column. Various chemical compositions have been reported describing an important chemical variability, and they have recently been reviewed [5]. The authors listed the 264 compounds that have been identified at least once in *C. mixtus* essential oil isolated from plants harvested all around the Mediterranean Sea (Algeria, Morocco, France, and Italy).

We focused our attention on Moroccan *C. mixtus* essential oil, which also displayed a fair chemical variability. According to the literature data, the reported essential oil compositions are distributed within two groups: (i) those that displayed a major component (Group 1) and (ii) those that contained various components with more or less similar content (Group 2).

- The compositions of essential oils of group 1 were dominated either by santolina alcohol (24.1–66.0%) [6–11], camphor (17.8–33.0%) [12–14], 2-methyl-2-trans-butenyl methacrylate (32.0–35.2%) [12,15,16], (E)-β-farnesene (35.5–50.3%) [12,15], or (E)-nerolidol (44.1%) [13];
- Samples of group 2 contained mainly monoterpenes (α-pinene, myrcene, 1,8-cineole, and camphor), irregular monoterpenes (santolina triene and santolina alcohol), sesquiterpene hydrocarbons (germacrene D and (E)-β-farnesene), or miscellaneous (2-tridecanone and (Z)-methyl isoeugenol) [12,13,17,18].

The aim of this paper that reports on the composition of two *C. mixtus* oil samples submitted to combined analysis by chromatographic and spectroscopic techniques is to demonstrate overall the importance of $^{13}$C NMR in identifying uncommon oxygenated sesquiterpenes, the presence of which induces an unusual composition.

## 2. Materials and Methods

### 2.1. Plant Material and Essential Oil Isolation

Aerial parts of *C. mixtus* have been collected in two locations (Figure 1, Table 1). Hydrodistillation (2 h) using a Clevenger-type apparatus of *C. mixtus* aerial parts (300 g) in 2 L flask yielded 0.3 mL of essential oil for both samples. To avoid any damage, the samples were stored at 5 °C in amber vials.

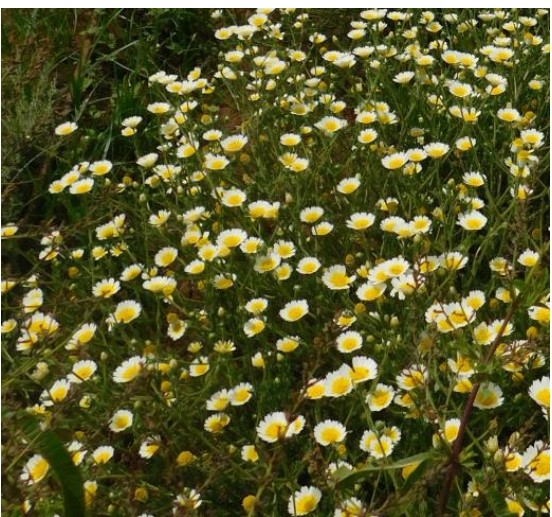

**Figure 1.** *Cladanthus mixtus* (L.) Chevall (Souk Had location).

**Table 1.** Characteristics of localities of harvest.

| Oil Sample | Location of Harvest | Elevation (m) | Longitude/Latitude | Date of Harvest |
|---|---|---|---|---|
| SH | Souk Had | 14 | N: 34°51′13.81″ W: 6°39′04.82″ | 6 April 2022 |
| SS | Ain Chkef forest Sidi Slimane | 451 | N: 33°98′57.60″ W: 5°01′84.39″ | 12 April 2022 |

*2.2. GC-FID Analysis*

GC-FID analyses were carried out using a Clarus 500 Perkin Elmer (Perkin Elmer, Courtaboeuf, France) chromatograph equipped with two FID and two fused-silica capillary columns (50 m length × 0.22 mm internal diameter, film thickness 0.25 μm), BP-1 (polydimethylsiloxane), and BP-20 (polyethylene glycol). The oven temperature was programmed from 60 °C to 220 °C at 2 °C/min and then held isothermal at 220 °C for 20 min; injector temperature, 250 °C; detector temperature, 250 °C; carrier gas, $H_2$ (0.8 mL/min); split, 1/60; and injected volume, 0.5 μL. The relative proportions of the essential oil constituents were expressed as percentages obtained by peak-area normalization without using correcting factors. Retention indices (RI) were determined relative to the retention times of a series of *n*-alkanes (C8–C28) with linear interpolation (Target Compounds software from Perkin Elmer, V1.2019, Courtaboeuf, France).

*2.3. GC/MS Analysis*

GC/MS analyses were performed on a Clarus SQ8S Perkin Elmer TurboMass detector (quadrupole), directly coupled to a Clarus 580 Perkin-Elmer Autosystem XL chromatograph, equipped with a BP-1 (polydimethylsiloxane) fused-silica capillary column (50 m length × 0.22 mm internal diameter, film thickness 0.25 μm). The oven temperature was programmed from 60 to 220 °C at 2°/min and then held isothermal at 220 °C for 30 min; injector temp., 250 °C; ion source temp., 150 °C; carrier gas, He (1 mL/min); split ratio, 1:80; injection volume, 0.5 μL; and ionization energy, 70 eV. The electron ionization (EI) mass spectra were acquired over the mass range of 35–350 Da.

*2.4. Nuclear Magnetic Resonance*

[13]C NMR spectra were recorded on a Bruker AVANCE 400 Fourier Transform spectrometer equipped with a 5 mm probe operating at 100.63 MHz for [13]C, in $CDCl_3$, with all shifts referred to internal Tetramethylsilane (TMS) at room temperature (298 °C). The following parameters were used: pulse width = 4 μs (flip angle 45°); acquisition time = 2.7 s for 128 K data table with a spectral width of 25,000 Hz (250 ppm); CPD mode decoupling; and digital resolution = 0.183 Hz/pt. For each sample (40 mg of essential oil in 0.5 mL of $CDCl_3$), 3000 scans were recorded.

*2.5. Identification of Individual Components*

Identification of the individual components was carried out (i) by comparison of their GC retention indices (RI) on non-polar and polar columns with those of reference compounds compiled in the in-house library and with literature data [19–21]; (ii) on computer matching against commercial mass spectral libraries [21–23]; and (iii) on comparison of the signals in the [13]C NMR spectra of the samples with those of reference spectra compiled in the laboratory spectral library, with the help of a laboratory-made software [24,25] (Figure 2).

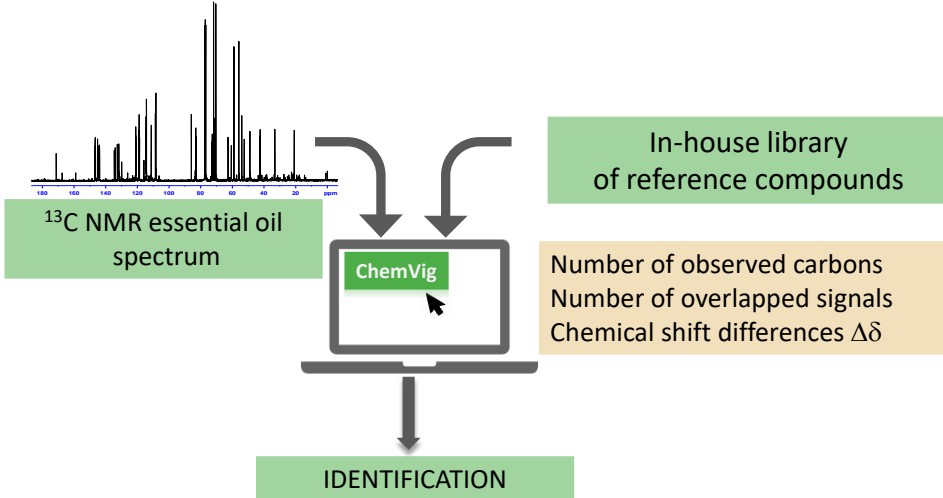

**Figure 2.** Identification of individual compounds using $^{13}$C NMR.

### 3. Results

*3.1. Methods for Identification of Individual Components of Essential Oils*

Identification of individual components of essential oils is routinely performed by using a fast-scanning mass spectrometer associated with a gas chromatograph. The mass spectrum of every component is compared with those of reference compounds compiled in commercial or in-house libraries. Two-dimensional GC coupled with MS may be useful for the analysis of complex essential oils in order to individualize components that are co-eluted when a unique column is employed and then to record more reliable mass spectra. Commercial libraries contain mass spectra of thousands of compounds (covering all fields of research); among these, a few thousand belong to volatile components of essential oils. In-house MS libraries are constructed with pure compounds or compounds whose identity is ascertained in one essential oil sample by spectroscopic techniques. Therefore, the reference MS spectra of various new compounds identified in essential oils and reported in the literature are not directly available to be introduced in a given MS library. In most analyses, identification of the compounds suggested by MS is confirmed by comparison of its retention indices (RI) on non-polar and polar chromatography columns with those of reference compounds compiled in the literature and/or homemade RI libraries.

In parallel, it has been shown that $^{13}$C NMR can be used for the non-destructive, non-separative identification of individual components of essential oils. In this computerized procedure developed at the University of Corsica, an individual component is identified by comparison of the signals of the mixture spectrum with those of reference spectra compiled in a library [24,25] (Figure 2). It should be pointed out that structural elucidation of every new compound proceeds, inter alia, via $^{13}$C NMR spectroscopy, and therefore, the $^{13}$C NMR spectrum is fully reported in the publication. It could be added that nowadays, a high-field spectrometer allows us to record the $^{13}$C NMR spectrum of isolated compounds at the mg level. The diluted solutions avoid intermolecular influence, and therefore, chemical shifts are perfectly reproducible. In practice, two spectral data libraries were constructed; the first one contains spectra recorded in the lab, and the second contains spectra reported in the literature for every new compound isolated from plants or obtained by synthesis. Both libraries are continuously implemented. Each component is identified considering three parameters directly available from the in-house computer program: (i) the number of observed carbons with respect to the number of expected signals, (ii) the number of overlapped signals of carbons that possess the same chemical shift, and (iii) the difference of the chemical shift of each signal in the mixture spectrum and in the reference (Figure 2). A compound is considered as identified when at least 50% of its signals belonging solely to that molecule are observed [24,25].

The benefit of using various chromatographic and spectroscopic techniques for the analysis of essential oil has been demonstrated and exemplified. For instance, the key role of [13]C NMR analysis in the identification of individual components of Ivoirian *Polyalthia longifolia* leaf oil and of *Xanthocyparis vietnamensis* wood oil has been highlighted [26,27]. [13]C NMR analysis also appeared suited for the identification of stereoisomers [28,29].

### 3.2. Chemical Composition of the Two Oil Samples

EOs were isolated from aerial parts of *C. mixtus* harvested in North-Western Morocco, at Souk Had (sample SH) and at Ain Chkef forest, near Sidi Slimane (Sample SS). Yields were measured as 0.10% (*v*/*w*) for both oil samples, which were submitted to GC(RI) on two column (non-polar and polar phases), GC/MS, and [13]C NMR analyses.

We will detail below the identification of individual components of two Moroccan *C. mixtus* essential oil samples (Table 2) by (i) GC(RI), GC/MS, and [13]C NMR (major components), (ii) GC(RI) and GC/MS (minor components), and (iii) by GC(RI) and [13]C NMR (uncommon oxygenated sesquiterpenes).

**Table 2.** Chemical composition of two oil samples from aerial parts of Moroccan *Cladanthus mixtus*.

| N° | Components [a] | RIa Lit [b] | RIp Lit [b] | RIa | RIp | SH% | SS% | Identification Mode |
|----|----|----|----|----|----|----|----|----|
| 1 | 3-Methyl-1-pentanol | 825 [c] | 1327 [c] | 829 | 1326 | - | 0.1 | RI, MS |
| 2 | Isobutyl isobutyrate | 899 | 1095 | 901 | 1096 | 0.9 | 0.1 | RI, MS, [13]C NMR |
| 3 | α-Thujene | 932 | 1025 | 924 | 1019 | 0.7 | 0.8 | RI, MS, [13]C NMR |
| 4 | α-Pinene | 936 | 1026 | 932 | 1019 | 16.1 | 4.6 | RI, MS, [13]C NMR |
| 5 | Camphene | 947 | 1068 | 945 | 1066 | 0.2 | 0.2 | RI, MS |
| 6 | Sabinene | 973 | 1122 | 967 | 1126 | 3.2 | 1.0 | RI, MS, [13]C NMR |
| 7 | β-Pinene | 978 | 1110 | 972 | 1115 | 0.5 | 0.1 | RI, MS |
| 8 | Myrcene | 987 | 1161 | 982 | 1164 | 1.1 | 1.3 | RI, MS, [13]C NMR |
| 9 | Isobutyl 2-methylbutyrate | 990 [d] | 1183 [d] | 990 | 1183 | 0.2 | - | RI, MS |
| 10 | Isobutyl isovalerate | 992 [d] | 1179 [d] | 992 | 1179 | 0.1 | - | RI, MS |
| 11 | Isopentyl isobutyrate | 996 [d] | 1195 [d] | 999 | 1195 | 0.1 | 0.1 | RI, MS |
| 12 | 2-Methylbutyl isobutyrate | 1003 [d] | 1201 [d] | 1003 | 1201 | 1.1 | - | RI, MS, [13]C NMR |
| 13 | α-Terpinene | 1011 | 1178 | 1010 | 1183 | - | 0.1 | RI, MS |
| 14 | *p*-Cymene | 1015 | 1270 | 1013 | 1276 | 2.2 | 0.2 | RI, MS, [13]C NMR |
| 15 | Limonene | 1025 | 1198 | 1022 | 1205 | 3.6 * | 1.0 * | RI, MS, [13]C NMR |
| 16 | 1,8-Cineole | 1025 | 1211 | 1022 | 1215 | 20.8 * | 3.9 * | RI, MS, [13]C NMR |
| 17 | Santolina alcohol | 1019 [e] | 1391 [e] | 1022 | 1404 | - | 3.4 * | RI, MS, [13]C NMR |
| 18 | Isobutyl angelate | 1036 [d] | 1293 [d] | 1035 | 1293 | 1.5 | - | RI, MS, [13]C NMR |
| 19 | γ-Terpinene | 1051 | 1245 | 1050 | 1249 | 0.3 | 0.2 | RI, MS |
| 20 | Artemisia alcohol | 1071 | 1510 | 1070 | 1507 | - | 0.2 | RI, MS |
| 21 | Nonanal | 1084 | 1391 | 1083 | 1389 | - | 0.1 | RI, MS |
| 22 | Linalool | 1086 | 1544 | 1084 | 1550 | 0.6 | 0.1 | RI, MS |
| 23 | Hotrienol | 1088 | 1602 | 1086 | 1611 | - | 0.6 | RI, MS, [13]C NMR |
| 24 | 2-Methylbutyl 2-methyl butyrate | 1089 [f] | 1279 [f] | 1090 | 1283 | 0.5 | - | RI, MS, [13]C NMR |
| 25 | 3-Methylpentyl isobutyrate | 1095 [g] | nd | 1103 | nd | 0.8 | 0.2 | RI, MS |
| 26 | α-Campholenal | 1107 | 1496 | 1107 | 1486 | - | 0.1 | RI, MS |
| 27 | trans-Pinocarveol | 1126 | 1661 | 1125 | 1655 | 0.7 | 0.4 | RI, MS, [13]C NMR |
| 28 | Pinocarvone | 1140 | 1575 | 1142 | 1569 | 0.2 | 0.5 | RI, MS |
| 29 | Borneol | 1153 | 1700 | 1152 | 1703 | 1.8 | 1.2 | RI, MS, [13]C NMR |
| 30 | Terpinen-4-ol | 1164 | 1601 | 1164 | 1605 | 3.5 | 0.5 | RI, MS, [13]C NMR |
| 31 | α-Terpineol | 1176 | 1694 | 1175 | 1699 | 2.1 | 0.3 | RI, MS, [13]C NMR |
| 32 | Myrtenol | 1182 | 1790 | 1180 | 1790 | - | 0.4 | RI, MS |
| 33 | Bornyl acetate | 1270 | 1580 | 1271 | 1583 | - | 0.1 | RI, MS |
| 34 | (Z)-2-Hexenyl hexanoate | 1333 [h] | 1653 [i] | 1327 | 1655 | - | 0.1 | RI, MS |
| 35 | δ-Elemene | 1340 | 1469 | 1335 | 1470 | - | 0.5 | RI, MS, [13]C NMR |
| 36 | 7α-Silphiperfol-5-ene | 1348 [j] | 1454 [j] | 1347 | 1452 | 0.4 | tr | RI, MS |
| 37 | Geranyl acetate | 1362 | 1752 | 1360 | 1742 | - | 0.1 | RI, MS |
| 38 | Cyclocopacamphene | 1368 | 1483 | 1362 | 1483 | 0.4 | 0.6 | RI, MS |
| 39 | α-Copaene | 1375 | 1491 | 1375 | 1491 | - | 0.3 | RI, MS |

Table 2. *Cont.*

| N° | Components [a] | RIa Lit [b] | RIp Lit [b] | RIa | RIp | SH% | SS% | Identification Mode |
|----|----|----|----|----|----|----|----|----|
| 40 | β-Elemene | 1388 | 1591 | 1389 | 1591 | 0.4 | 0.9 | RI, MS, $^{13}$C NMR |
| 41 | Bornyl isobutyrate | 1402 [k] | 1641 [k] | 1402 | 1643 | - | 0.1 | RI, MS |
| 42 | (E)-β-Caryophyllene | 1419 | 1598 | 1416 | 1595 | - | 0.9 | RI, MS |
| 43 | (E)-β-Farnesene | 1446 | 1664 | 1446 | 1667 | 0.8 | 2.2 | RI, MS, $^{13}$C NMR |
| 44 | 3,6,6,9-bis-epoxy-Farnesa-1,7(14),10-triene | 1450 [l] | 1831 [l] | 1449 | 1828 | 4.3 * | 5.8 * | RI, $^{13}$C NMR |
| 45 | 9-epi-3,6,6,9-bis-epoxy Farnesa-1,7(14),10-triene | 1450 [l] | 1834 [l] | 1449 | 1831 | 0.9 * | 0.9 * | RI, $^{13}$C NMR |
| 46 | 3,9-di-epi-3,6,6,9-bis-epoxy-Farnesa-1,7(14),10-triene | 1458 [l] | 1865 [l] | 1457 | 1867 | 0.9 * | 1.9 * | RI, $^{13}$C NMR |
| 47 | 3 epi-3,6,6,9-bis-epoxy-Farnesa-1,7(14),10-triene | 1458 [l] | 1870 [l] | 1457 | 1868 | 2.2 * | 1.1 * | RI, $^{13}$C NMR |
| 48 | Selina-4,11-diene | 1474 [m] | 1670 [m] | 1470 | 1674 | - | 0.7 | RI, MS, $^{13}$C NMR |
| 49 | Germacrene D | 1476 | 1708 | 1478 | 1709 | 0.3 | 4.0 | RI, MS, $^{13}$C NMR |
| 50 | β-Selinene | 1481 | 1717 | 1481 | 1716 | - | 0.4 | RI, MS |
| 51 | (Z,E)-α-Farnesene | 1481 | 1728 | 1486 | 1714 | - | 0.2 | RI, MS |
| 52 | Bicyclogermacrene | 1490 | 1734 | 1490 | 1726 | - | 0.2 | RI, MS |
| 53 | α-Muurolene | 1491 | 1723 | 1492 | 1724 | 0.7 | 0.5 | RI, MS, $^{13}$C NMR |
| 54 | (E,E)-α-Farnesene | 1496 | 1744 | 1495 | 1749 | - | 0.2 | RI, MS |
| 55 | γ-Cadinene | 1506 | 1763 | 1505 | 1755 | - | 0.2 | RI, MS |
| 56 | δ-Cadinene | 1520 | 1756 | 1516 | 1757 | 0.2 | 0.6 | RI, MS, $^{13}$C NMR |
| 57 | β-Elemol | 1537 | 2088 | 1533 | 2053 | - | 0.7 | RI, MS |
| 58 | (E)-Nerolidol | 1550 | 2036 | 1550 | 2042 | 3.4 | 13.9 | RI, MS, $^{13}$C NMR |
| 59 | Spathulenol | 1566 | 2126 | 1563 | 2119 | - | 0.7 | RI, MS |
| 60 | Caryophyllene oxide | 1570 | 1986 | 1572 | 1983 | 1.7 | 0.9 | RI, MS, $^{13}$C NMR |
| 61 | Fokienol | 1577 [n] | 2170 [n] | 1577 | 2168 | 1.1 | 2.9 | RI, MS, $^{13}$C NMR |
| 62 | β-Oplopenone | 1593 | 2084 | 1588 | 2077 | - | 0.3 | RI, MS |
| 63 | Copaborneol | 1600 [k] | 2181 [k] | 1595 | 2176 | 1.3 | 1.0 | RI, MS, $^{13}$C NMR |
| 64 | 6,9-epoxy-Farnesa-1,7(14),10-trien-3-ol | 1601 [l] | 2257 [l] | 1598 | 2254 | 2.7 | 5.4 | RI, $^{13}$C NMR |
| 65 | 3-epi-6,9-epoxy-Farnesa-1,7(14),10-trien-3-ol | 1601 [l] | 2255 [l] | 1598 | 2249 | 1.1 | 4.3 | RI, $^{13}$C NMR |
| 66 | Junenol | 1607 [o] | 2055 [o] | 1607 | 2046 | - | 0.6 | RI, MS |
| 67 | 6-epi-6,9-epoxy-Farnesa-1,7(14),10-trien-3-ol | 1614l | 2276 [l] | 1610 | 2270 | 0.9 | 2.3 | RI, $^{13}$C NMR |
| 68 | 3,6-diepi-6,9-epoxy-Farnesa-1,7(14),10-trien-3-ol | 1614 [l] | 2284 [l] | 1612 | 2279 | 1.2 | 3.3 | RI, $^{13}$C NMR |
| 69 | epi-γ-Eudesmol | 1608 | 2106 | 1614 | 2107 | - | 0.4 | RI, MS |
| 70 | γ-Eudesmol | 1616 | 2176 | 1614 | 2186 | - | 0.5 | RI, MS |
| 71 | Alismol | 1620 [l] | 2295 [l] | 1617 | 2291 | 0.8 | 0.4 | RI, $^{13}$C NMR |
| 72 | Caryophylla-4(12),8(13)-dien-5α-ol | 1622 [l] | 2294 [l] | 1619 | 2291 | - | 0.5 | RI, MS |
| 73 | τ-Muurolol | 1631 | 2186 | 1625 | 2185 | 1.6 | 1.0 | RI, MS, $^{13}$C NMR |
| 74 | Torreyol (δ-Cadinol) | 1631 [p] | 2167 [p] | 1629 | 2164 | - | 0.2 | RI, MS |
| 75 | α-Cadinol | 1643 | 2227 | 1637 | 2226 | - | 1.1 | RI, MS, $^{13}$C NMR |
| 76 | α-Bisabolol | 1668 | 2213 | 1665 | 2201 | - | 0.8 * | RI, MS, $^{13}$C NMR |
| 77 | epi-α-Bisabolol | 1674 | 2214 | 1665 | 2212 | - | 0.8 * | RI, MS, $^{13}$C NMR |
| 78 | Shyobunol | 1687 [q] | 1953 [r] | 1675 | 1940 | - | 0.8 | RI, MS |
| 79 | Neophytadiene | 1827 [s] | 1933 [s] | 1835 | 1936 | - | 0.1 | RI, MS |
| 80 | (Z)-Phytol | 2077 [t] | 2551 [t] | 2096 | 2558 | - | 0.3 | RI, MS |
| 81 | (E)-Phytol | 2103 [t] | 2613 [t] | 2106 | 2609 | - | 0.2 | RI, MS |
| | Total | | | | | 90.1 | 87.6 | |

[a] Components are listed following their order of elution on non-polar column BP-1; percentages on non-polar capillary column, except those with * (%) on polar capillary column BP-20). RIa, RIp: Retention indices on non-polar and polar columns, respectively; $^{13}$C NMR: compound identified by $^{13}$C-NMR, at least in one oil sample; nd: not determined; and tr: traces. [b] RIa lit, RIp lit: literature retention indices [19] otherwise stated. [c] [30]; [d] [31]; [e] [32]; [f] [33]; [g] [20]; [h] [34]; [i] [35]; [j] [36]; [k] [37]; [l] [38]; [m] [39]; [n] [40]; [o] [41]; [p] [42]; [q] [43]; [r] [44]; and [s] [45]. [t] RIs of pure compounds from Aldrich (Z/E mixture).

In total, 81 compounds were identified. They accounted for 90.1 and 87.6% of the whole composition, respectively (Table 2). The chromatogram of the SS sample (non-polar column) is reported in Figure 3.

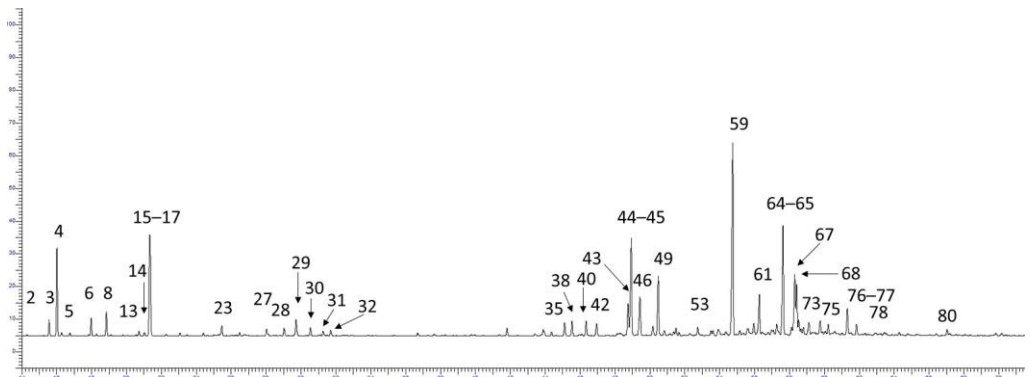

**Figure 3.** Chromatogram of SS sample (BP-1, non-polar column).

The identification of individual components was conducted as follows:

- Mass spectrometry in combination with retention indices on two capillary columns (non-polar and polar phases) allowed the identification of 73 components (**1–43**, **48–63**, **66**, **69–81**) from traces accounting for more than 21%;
- In parallel, the identification of all the major components of both oil samples was ascertained by [13]C NMR following the computerized methodology developed at the University of Corsica (**2–4**, **6**, **8**, **12**, **14–18**, **23**, **24**, **27**, **29–31**, **35**, **40**, **43**, **48**, **49**, **53**, **56**, **58**, **60**, **61**, **63**, **73**, and **75–77**) [25,26]. Alismol **71** was identified by [13]C NMR and RIs in both oil samples. Similarly, α-bisabolol **76** and epi-α-bisabolol **77** co-eluted on the non-polar column and were differentiated on the polar column. The occurrence of both epimers was confirmed by the observation of characteristic signals in the [13]C NMR spectra;
- Lastly, eight compounds (**44–47**, **64**, **65**, **67**, and **68**) that accounted for 0.2–8.3% each (percentages measured on the polar column due to overlapped GC signals on the non-polar column) remained unidentified regardless of the matching vs. MS commercial and homemade libraries at our disposal. Their retention indices were as follows: non-polar/polar column = 1451/1831, 1451/1834, 1460/1870, and 1460/1871 on the one hand, and 1600/2252, 1600/2255, 1613/2274, and 1613/2283 on the other hand. In parallel, computer alignment with the internal [13]C NMR library allowed the presence of eight components in significant content. Components **44–47** were identified as 3,6,6,9-bis-epoxy-farnesa-1,7(14),10-triene (IUPAC nomenclature (2S,5S,7S)-2-methyl-9-methylene-7-(2-methylprop-1-en-1-yl)-2-vinyl-1,6-dioxaspiro [4.4]nonane, relative stereochemistry)), and its epimers, 3-epi (2R,5S,7S), 9-epi (2S,5S,7R), and 3,9-diepi (2R,5S,7R). Components **64**, **65**, **67**, and **68** were identified as 6,9-epoxy-farnesa-1,7(14),10-trien-3-ol (IUPAC nomenclature (2S,3′R/S,5S)-(3′-hydroxy-3′-methylpent-4′-en)-3-methylene-5-isopropylidene tetrahydrofuran, relative stereochemistry)) and its epimers, 3-epi (2S, 3′R/S, 5S), 6-epi (2S, 3′R/S, 5R), and 3,6-diepi (2R, 3′R/S,5R) (Table 2, Figure 4).

The eight components were identified by comparison of their [13]C NMR chemical shifts, measured in the recorded [13]C NMR spectra of both oil samples with those of reference spectra compiled in our library (Figure 5, Table S1).

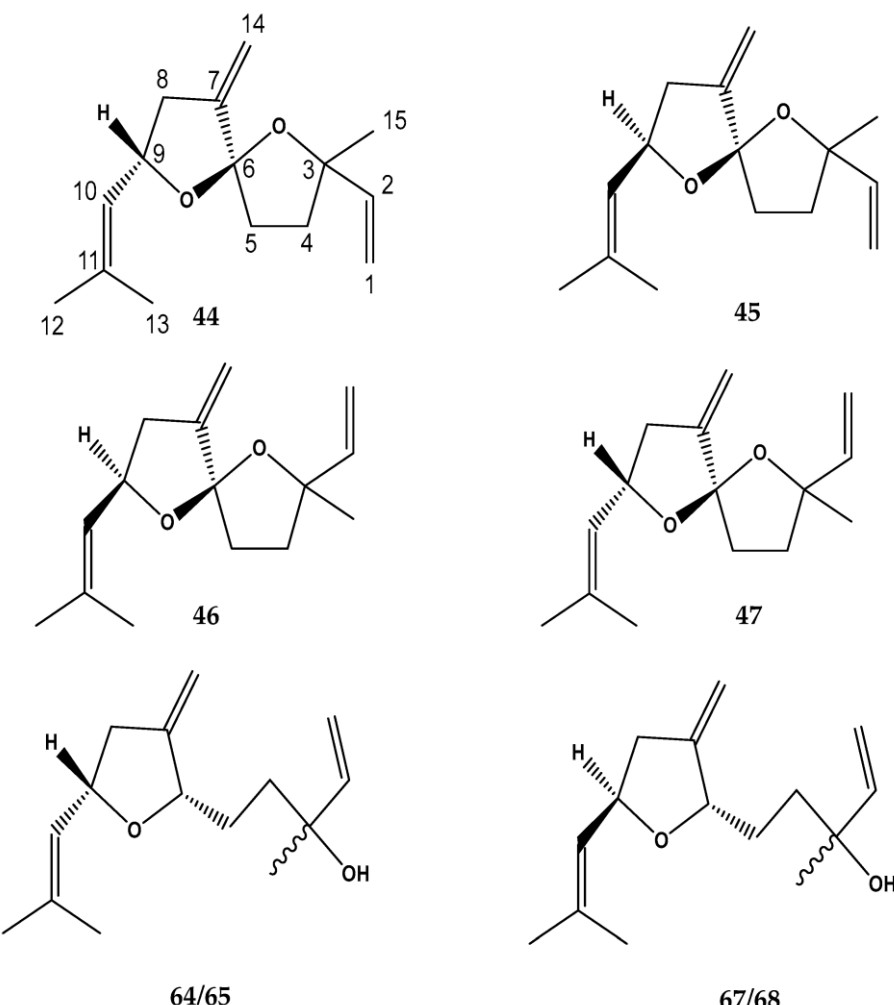

**Figure 4.** Structures of 3,6,6,9-bis-epoxy-farnesa-1,7(14),10-triene 44 and its epimers, 3-epi, 9-epi, and 3,9-diepi, and 6,9-epoxy-farnesa-1,7(14),10-trien-3-ol 64 and its epimers, 3-epi, 6-epi, and 3,6-diepi.

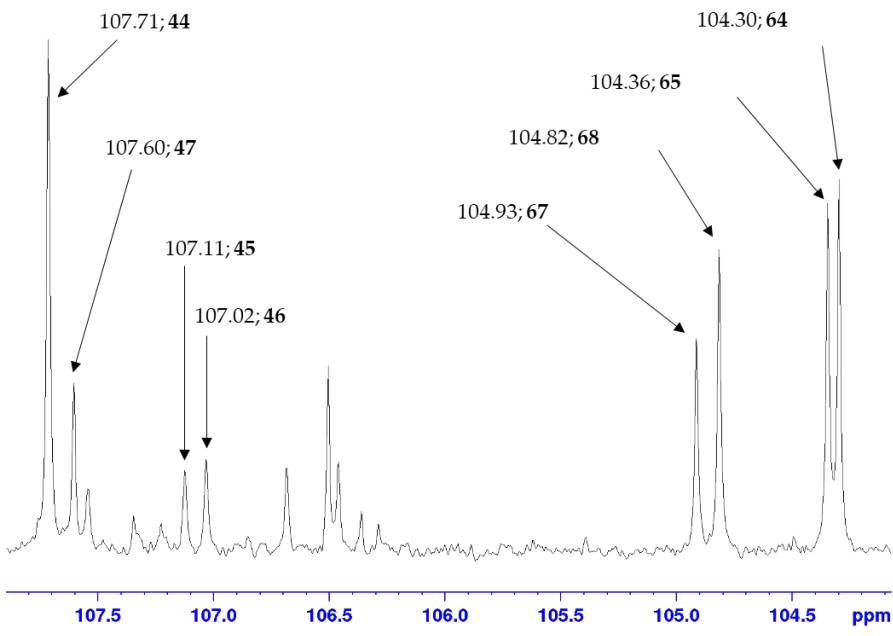

**Figure 5.** Part (104–108 ppm) of $^{13}$C-NMR spectrum of *C. mixtus* (C14, Csp2, methylene).

## 4. Discussion

Despite the similarity of the structures and the low percentage of some components, the chemical shifts of all carbons were observed, except those of the quaternary carbons of the minor isomers. The difference in chemical shifts between the experimental spectra and the reference data was always acceptable, as well as the number of overlapped signals. It could be noticed that the eight components were first isolated from the essential oil of *Tanacetum fruticulosum* and spectroscopically characterized by Weyerstahl et al., who reported, inter alia, their $^{13}$C NMR data [46]. To the best of our knowledge, since that time, the eight compounds have only been identified, by using $^{13}$C NMR spectroscopy, in essential oil isolated from aerial parts of Corsican *Dittrichia viscosa* [38].

The composition of the SH oil sample was dominated by monoterpenes, 1,8-cineole (20.8%), and α-pinene (16.1%), followed by limonene (3.6%), terpinen-4-ol (3.5%), sabinene (3.2%), and α-terpineol (2.1%). Sample SS contained mainly (E)-nerolidol (13.9%), besides α-pinene (4.6%) and santolina alcohol (3.4%). A few compounds were present at appreciable content in both oil samples: myrcene (1.1/1.3%), 3-methylpentyl isobutyrate (0.8/0.2%), borneol (1.8/1.2%), caryophyllene oxide (1.7/0.9%), fokienol (1.1/2.9%), copaborneol (1.3/1.0%), and τ-muurolol (1.6/1.0%). Some components were noticeable in the SH oil sample, as follows: isobutyl isobutyrate (0.9%), α-thujene (0.7%), β-pinene (0.5%), 2-methylbutyl isobutyrate (1.1%), p-cymene (2.2%), isobutyl angelate (1.5%), linalool (0.6%), and 7α-silphiperfol-5-ene (0.4%). Components in the SS oil sample were as follows: santolina alcohol (3.4%), β-elemene (0.9%), (E)-β-caryophyllene (0.9%), and (E)-β-farnesene (2.2%). It is noticeable that irregular monoterpene alcohols, santolina alcohol (3.4%) and artemisa alcohol (0.2%), and diterpenes, neophytadiene (0.1%), (Z)-phytol (0.3%), and (E)-phytol (0.2%) were found only in SS sample. Various hemiterpene esters and analogs usually found in *Cladanthus* species [31] (up to 1.5%) were identified: isobutyl isobutyrate, isobutyl 2-methylbutyrate, isobutyl isovalerate, isopentyl isobutyrate, 2-methylbutyl isobutyrate, isobutyl angelate, 2-methylbutyl 2-methyl butyrate, 3-methylpentyl isobutyrate, and (Z)-2-hexenyl hexanoate. Lastly, it could be pointed out that the content of 3,6,6,9-bisepoxyfarnesa-1,7(14),10-triene (**44**) and its three epimers (**45**, **46**, and **47**) were similar in both oil samples (8.3% vs. 9.7%), while the content of 6,9-epoxy-farnesa-1,7(14),10-trien-3-ol (**64**) and its epimers (**65**, **67**, and **68**) was substantially higher in sample SS than in sample SH (15.3% vs. 5.9%).

## 5. Conclusions

Although oil samples whose chemical composition was dominated by (i) α-pinene and/or 1,8-cineole, (ii) (E)-nerolidol, or (iii) (E)-β-farnesene have been reported, the occurrence of oxygenated farnesane derivatives at appreciable contents (up to 25%) brings originality to the investigated *C. mixtus* oil samples. Our results confirm the tremendous chemical variability of Moroccan *C. mixtus* essential oil and the potential of $^{13}$C NMR analysis, in combination with GC(RI), for the identification of uncommon oxygenated sesquiterpenes.

**Supplementary Materials:** The following supporting information can be downloaded at https://www.mdpi.com/article/10.3390/compounds3020028/s1. Table S1: $^{13}$C NMR chemical shifts of compounds **44–47**, **64**, **65**, **67**, and **68**.

**Author Contributions:** Conceptualization, K.B. and M.O.; methodology, K.B., M.P. and J.C.; validation, S.E.H. and K.B.; formal analysis, M.P. and J.C. investigation, S.E.H.; resources, M.O. and A.B.; writing—original draft preparation, K.B. and J.C.; writing—review and editing, J.C. and F.T.; visualization, K.B., F.T. and J.C.; project administration, K.B. All authors have read and agreed to the published version of the manuscript.

**Funding:** This research received no external funding.

**Data Availability Statement:** Data are contained within the article.

**Acknowledgments:** We thank all the farmers from the SH and SS locations for their kindness and help.

**Conflicts of Interest:** The authors declare no conflict of interest.

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
