# Peer review of "Integrated Analysis by GC/MS and 13C NMR of Moroccan Cladanthus mixtus Essential Oil; Identification of Uncommon Epoxyfarnesanes"

_compounds, doi:10.3390/compounds3020028_

Round 1
Reviewer 1 Report
Manuscript Number: compounds-2367936
Entitled: Integrated analysis by GC/MS and 13C NMR of Moroccan Cladanthus mixtus essential oil. Identification of uncommon epoxyfarnesanes
This is an interesting scientific study. Therefore, the manuscript is suitable for Compounds after considering the below comments:
- Title. Please modify the title. There should not be two sentences inside the title. Please remove dot.
- Please use the UIPAC names, e.g., (7S)-2-methyl-9-methylene-7-(2-methylprop-1-en-1-yl)-2-vinyl-1,6-dioxaspiro[4.4]nonane instead of “4”. Figure 2 should be redrawn, and every structure should be named. Below the structure could be, e.g., “4”, but in the text, the UIPAC name. Please check the correctness of all the structures.
- Cladanthus mixtus (L.) Chevall, please add it’s a photo.
- Do I understand correctly? Relying only on the GC-MS technique, the authors identified compounds, and the 13C NMR technique was used only to confirm the presence of selected molecules. Did the authors identify the components of each essential oil only based on both techniques? Please explain your idea and prove your thesis. For now, I doubt the ability to identify a mixture composed of 81 essential oil components based only on 1D 13C NMR, even if the authors have their own library of 13C NMR spectra of pure compounds. An additional complication is the fact that each of the 81 ingredients is in different concentrations. Additionally, each carbon possesses its own relaxation time and, consequently, different intensity in 13C NMR spectra. Please provide a chromatogram of the test mixtures. Please explain your reasoning.
Author Response
Dear Editor,
We wish to submit a revision of the manuscript. We considered all the suggestions and requirements of reviewers. We revised the manuscript accordingly with all changes listed below.
We hope that this manuscript is well suited for publication in Compounds.
Editor
The article should report scientifically sound experiments and provide a substantial amount of new information, and the suggested minimum word count is 4000 words (exclude author information and references). So, we kindly suggest you enrich your article to at least 4000 words.
As suggested, the number of words is increased.
Some sentences are detected as similar or the same with other papers. They are highlighted in the file attached. However, in case of any copyright controversy, we kindly ask you to make necessary revisions on these content. Please understand that more than 7 continuous duplicate words should be avoided.
We agree. Few sentences are similar or the same with others papers. They concern mainly the paragraph 2 “Materials and methods” in which the experimental parameters usually used by our team are reported.
In the whole text, several corrections are done to avoid 7 continuous duplicate words.
Referee 1
- Title. Please modify the title. There should not be two sentences inside the title. Please remove dot.
We remove the dot.
- Please use the UIPAC names, e.g., (7S)-2-methyl-9-methylene-7-(2-methylprop-1-en-1-yl)-2-vinyl-1,6-dioxaspiro[4.4]nonane instead of “4”. Figure 2 should be redrawn, and every structure should be named. Below the structure could be, e.g., “4”, but in the text, the UIPAC name. Please check the correctness of all the structures.
We included the IUPAC name of eight compounds in the text. Figure 2 (now figure 4) has been modified.
However, the IUPAC nomenclature is not usually used in essential oils. Indeed, the traditional nomenclature used in the terpene field are derived from the genus or family names of the plants. Then, to homogenize the names with the other 73 compounds, we chose to use the traditional nomenclature (farnesane skeleton).
- Cladanthus mixtus (L.) Chevall, please add it’s a photo.
A photo is added (Figure 1). This photo was also present in the graphical abstract.
- Do I understand correctly? Relying only on the GC-MS technique, the authors identified compounds, and the 13C NMR technique was used only to confirm the presence of selected molecules. Did the authors identify the components of each essential oil only based on both techniques? Please explain your idea and prove your thesis
GC/MS and 13C NMR technique were used in parallel. Seventy-three components were identified by GC/MS, 41 by 13C NMR. Among these, 39 components were identified only by GC/MS (and RIs), 32 by GC/MS and 13C NMR, and 9 by 13C NMR (and RIs).
For now, I doubt the ability to identify a mixture composed of 81 essential oil components based only on 1D 13C NMR, even if the authors have their own library of 13C NMR spectra of pure compounds.
Following the pioneering work of V. Formácek and and K.H. Kubeczka. (1982) »13C NMR analysis of essential oils in aromatic plants: basic and applied aspects », the identification of individual components of essential oil using 13C NMR spectroscopy without isolation or purification, is recognized as an analytical technique. In our lab, we performed a thousand of analyses. The simultaneous identification of various components by GC/MS and 13C NMR confirmed the validity of the method.
An additional complication is the fact that each of the 81 ingredients is in different concentrations. Additionally, each carbon possesses its own relaxation time and, consequently, different intensity in 13C NMR spectra.
All the components belong to the terpene family. The relaxation time of carbon atoms (quaternary carbons excepted) exhibits close values. Anyway, we don’t perform quantitative analysis.
Please provide a chromatogram of the test mixtures. Please explain your reasoning;
We provide a chromatogram and develop the reasoning (lines 194-215).
Referee 2
- Compound 44, mentioned in Figure 3, does not appear in the paper. Please recheck the compound number.
There is a mistake in figure 3. The compound number is 44 instead of 4.
- The paper did not compare the 13C NMR data obtained with the data from the cited literature. (Figure 1)
We agree. The eight components have been identified by comparison with Chemical shifts (13C NMR) compiled in our in-house library. Figure 2 is modified accordingly.
Some literature in the manuscript is marked in red, while others are not marked in red. Please unify.
Done.
- Why is Altitude (m) highlighted in red in Table 1?
Done. We propose elevation instead of altitude
- References should be hyperlinked uniformly, please follow the journal format requirements.
Done.
- There should be a space between the number and unit, for example, "0.25μm" in line 86 of the text, please correct it.
Done.
- Please correct the subscript of CDCl3 in line 107.
Done.
- Copies of GC-FID and GC/MS should be provide
We add a figure with a chromatogram on non-polar column.

Reviewer 2 Report
This paper presents a combined analysis of two essential oil samples from C. mixtus by chromatographic, spectroscopic techniques, and Nuclear Magnetic Resonance. The paper first introduced the basic situation of Moroccan Cladanthus mixtus, and then two essential oil samples have been isolated from aerial parts and analyzed by a combination of chromatographic and spectroscopic techniques [gas chromatography (GC) in combination with retention indices (RI), gas chromatography-mass spectrometry (GC/MS) and 13C NMR spectroscopy] to illustrate the relative proportions of the essential oil components. A total of 81 compounds were identified in this paper, and the identification of all major components of the two oil samples was determined by 13C NMR essential oil spectroscopy, and two spectral databases (home library and literature library). This paper concludes the tremendous chemical variability of Moroccan C. mixtus essential oil and the usefulness of 13C NMR analysis, in combination with GC(RI), for the identification of uncommon oxygenated sesquiterpenes. In general, this manuscript is of great significance for the study of essential oil composition. It is suggested to publish on Compounds after minor revisions.
Questions:
1. Compound 44, mentioned in Figure 3, does not appear in the paper. Please recheck the compound number.
2. The paper did not compare the 13C NMR data obtained with the data from the cited literature. (Figure 1)
3. Some literature in the manuscript is marked in red, while others are not marked in red. Please unify.
4. Why is Altitude (m) highlighted in red in Table 1?
5. References should be hyperlinked uniformly, please follow the journal format requirements.
6. There should be a space between the number and unit, for example, "0.25μm" in line 86 of the text, please correct it.
7. Please correct the subscript of CDCl3 in line 107.
8. Copies of GC-FID and GC/MS should be provided.
Author Response

(The authors gave the same response as above.)

Round 2
Reviewer 1 Report
Manuscript Number: compounds-2367936
Entitled: Integrated analysis by GC/MS and 13C NMR of Moroccan Cladanthus mixtus essential oil; identification of uncommon epoxyfarnesanes.
This manuscript needs below corrections:
- Title. Please remove dot. The authors did not correct it.
- Page 2; “were stored at 5add space°C in”, and passim.
- The authors, in Figure, presented a chromatogram; however, not every signal is identified. Please explain why not identifying signals between 4 and 15-17 or before 43 have no impact on 13C NMR? Please explain the differences between carbon atoms of all presented in Fig. 4 molecules. You can show a compilation of several 13C NMR spectra. Not every signal is identified in Fig. 5, 13C NMR spectrum, even in such a small range.
I am against the publication of this manuscript until the authors have accurately stated their ideas and results.
Author Response
Dear Editor,
We wish to submit a revision of the manuscript. We considered all the suggestions and requirements of referee. We revised the manuscript accordingly with all changes listed below.
- Title. Please remove dot. The authors did not correct it.
Done
- Page 2; “were stored at 5add space°C in”, and passim.
Done
- The authors, in Figure, presented a chromatogram; however, not every signal is identified.
In figure 3, chromatogram of the sample SS, only the major components were mentioned, as this is usually done. Nevertheless, in response to the referee, we mentioned other components present at appreciable content.
Please explain why not identifying signals between 4 and 15-17 or before 43 have no impact on 13C NMR?
All the components identified by 13C NMR were observed on the chromatogram through their RIs, and quantified (Table 2). Otherwise, it appears that 32 out of 41 components identified by NMR have been also identified by GC/MS, that validate the identification of individual components in an essential oil sample up to 0.5%.
Please explain the differences between carbon atoms of all presented in Fig. 4 molecules. You can show a compilation of several 13C NMR spectra.
The four isomers 44-47, which are diastereoisomers, exhibit a different series of 13C NMR chemical shifts. Then, 13C NMR is an efficient method to identify these compounds. We added the above table as supplementary material (Table S1).
|
44 |
45 |
46 |
47 |
|
149.84 |
149.41 |
150.05 |
150.55 |
|
143.86 |
143.74 |
144.57 |
144.78 |
|
137.86 |
134.62 |
134.20 |
137.88 |
|
124.79 |
126.73 |
126.71 |
124.73 |
|
113.03 |
113.07 |
113.02 |
113.09 |
|
111.30 |
111.49 |
111.75 |
111.36 |
|
107.71 |
107.11 |
107.02 |
107.60 |
|
83.98 |
84.23 |
84.04 |
83.85 |
|
72.22 |
72.97 |
73.12 |
72.52 |
|
38.63 |
38.36 |
38.26 |
38.70 |
|
36.90 |
36.55 |
37.05 |
37.20 |
|
35.74 |
35.86 |
36.47 |
36.25 |
|
28.41 |
28.11 |
25.91 |
26.23 |
|
25.91 |
25.78 |
25.74 |
25.92 |
|
18.33 |
18.08 |
18.01 |
18.32 |
It is noteworthy that isomers 64, 65, 67 and 68 also show different series of 13C NMR chemical shifts.
|
64 |
65 |
67 |
68 |
|
151.62 |
151.57 |
151.69 |
151.75 |
|
145.17 |
145.12 |
145.21 |
145.26 |
|
137.08 |
137.04 |
136.50 |
136.43 |
|
124.68 |
124.78 |
124.87 |
125.09 |
|
111.67 |
111.75 |
111.69 |
111.66 |
|
104.30 |
104.36 |
104.93 |
104.82 |
|
80.60 |
80.85 |
80.34 |
79.79 |
|
75.06 |
75.07 |
73.65 |
73.72 |
|
72.66 |
72.70 |
72.70 |
72.74 |
|
40.21 |
40.04 |
39.70 |
39.72 |
|
37.37 |
37.94 |
38.83 |
38.03 |
|
29.07 |
29.67 |
29.98 |
29.47 |
|
28.17 |
28.42 |
28.11 |
28.49 |
|
25.81 |
25.80 |
25.79 |
25.82 |
|
18.35 |
18.33 |
18.23 |
18.24 |
Not every signal is identified in Fig. 5, 13C NMR spectrum, even in such a small range.
The signals with chemical shits present in the zoom around 106 ppm belong to minor oxygenated sesquiterpenes (alismol, spathulenol, and/or unidentified components…). For instance, the signal at 56.51 ppm belongs to alismol. We preferred to focus our attention on the signals of eight epoxy/bis-epoxy-farnesanes 44-47, 64, 65, 67 and 68.
Best regards,
Félix Tomi

Round 3
Reviewer 1 Report
Dear Authors,
Thank you so much for your kind letter.
As an organic chemist, I know that identifying anything from reaction mixtures is a huge challenge. Sometimes, we can use NMR techniques to identify compounds from a mixture of molecules. We can choose diagnostic signals if compounds have such in such a situation. The author does not mention such a situation.
The chemical shift in NMR spectroscopy is influenced by factors such as concentration, viscosity, and temperature. This also means that different concentrations of the compounds will also affect the appearance of 13C NMR spectra.
https://www.slideshare.net/TaherPatel1/factors-affecting-chemical-shift-82948401
The 31P NMR or 19F NMR is a more diagnostic technique if your mixture of compounds has only one molecule that possesses phosphorus or fluoride atom.
Please change your phrase:
„Computer matching against in-house 13C NMR library allowed the identification of the eight components at appreciable contents.”
Please change to
„Computer alignment with the internal 13C NMR library allowed the presence of eight components in significant content to be confirmed.”
Dear Authors, please do not use the word „identification,” in the manuscript text for 13C NMR spectroscopy.
The essential oil could have a different composition of molecules in one type of plant. The factors affecting it are lighting, soil type, the amount of water or fertilizers, and so on.
In conclusion, please add some explanation for the readers using my above advice.
Author Response
As suggested, we modify the sentence line 210.
Computer alignment with the internal 13C NMR library allowed the presence of eight components in significant content.
instead of
Computer matching against in-house 13C NMR library allowed the identification of the eight components at appreciable contents.
However, we don’t replace the word “identification” by “presence” in the whole text. Indeed, it is clearly an identification.
https://www.researchgate.net/profile/Felix-Tomi. For more than thirty years, more than two hundred articles described and used the methodology of identification of volatile compounds (terpenes, phenylpropanoids, acyclic…) using 13C NMR spectroscopy. The term “identification by 13C NMR spectroscopy” is used in all articles and has been validated by all the referees.